# Benefits of Combining *Sonchus brachyotus DC.* Extracts and Synbiotics in Alleviating Non-Alcoholic Fatty Liver Disease

**DOI:** 10.3390/foods12183393

**Published:** 2023-09-11

**Authors:** Wenwu Huang, Boyuan Shen, Xiumei Li, Tongcun Zhang, Xiang Zhou

**Affiliations:** 1College of Life Sciences & Health, Wuhan University of Science & Technology, Wuhan 430065, China; wenwu@wust.edu.cn (W.H.); shenboyuan07@163.com (B.S.); zhangtongcun@wust.edu.cn (T.Z.); 2Key Laboratory of Feed Biotechnology, Ministry of Agriculture and Rural Affairs, Institute of Feed Research of CAAS, Beijing 100000, China; lixiumei@caas.cn

**Keywords:** NAFLD, synbiotics, gut microbiota, *Sonchus brachyotus DC.* extracts

## Abstract

Non-alcoholic fatty liver disease, commonly abbreviated to NAFLD, is a pervasive ailment within the digestive system, exhibiting a rising prevalence, and impacting individuals at increasingly younger ages. Those afflicted by NAFLD face a heightened vulnerability to the onset of profound liver fibrosis, cardiovascular complications, and malignancies. Currently, NAFLD poses a significant threat to human health, and there is no approved therapeutic treatment for it. Recent studies have shown that synbiotics, which regulate intestinal microecology, can positively impact glucolipid metabolism, and improve NAFLD-related indicators. *Sonchus brachyotus DC.*, a Chinese herb, exhibits hepatoprotective and potent antioxidant properties, suggesting its potential therapeutic use in NAFLD. Our preclinical animal model investigation suggests that the synergy between *Sonchus brachyotus DC.* extracts and synbiotics is significantly more effective in preventing and treating NAFLD, compared to the isolated use of either component. As a result, this combination holds the potential to introduce a fresh and encouraging therapeutic approach to addressing NAFLD.

## 1. Introduction

NAFLD is a clinicopathological condition characterized by hepatic steatosis and lipid accumulation. It serves as a prominent underlying factor for chronic liver diseases, carrying a notable risk of liver fibrosis, which can occur in up to 25% of cases [1]. NAFLD, often correlated with metabolic syndrome and type 2 diabetes mellitus (T2DM), fosters the progression of cirrhosis, hepatocellular carcinoma, cardiovascular disease, and extrahepatic cancers. Consequently, cardiovascular ailments, malignant tumors, and advanced liver conditions are the chief contributors to mortality in individuals afflicted by NAFLD.

In recent years, the prevalence of NAFLD has continued to rise, due to improved material living standards and dietary changes. Presently, non-alcoholic fatty liver disease (NAFLD) stands as one of the most widespread chronic liver conditions globally. It affects roughly 30% of adults, constituting a substantial fraction of the hepatic-related morbidity and mortality [2]. In 2018, the global prevalence of NAFLD was estimated at 25%, with 27.37% of cases in Asia. Modeling studies project a continuous 30% increase in fatty liver disease worldwide by 2030 compared to previous years in eight countries, including China, the UK, France, and Germany [3,4]. This alarming trend, with the NAFLD prevalence in China rising from 13% to 43% over the last 20 years [5], signifies a growing burden on individuals and society. Currently, there are no approved therapies for NAFLD, except for weight loss through various interventions to slow the disease progression. Current treatments focus on improving insulin sensitivity, and reducing liver enzyme levels through lifestyle changes and medications. However, these interventions lack comprehensiveness and effectiveness in alleviating NAFLD. The side effects and long-term administration of insulin sensitizers and vitamin E may limit their widespread acceptance [6]. Although dietary intervention and exercise are recommended as first-line therapies, they are often required in combination with pharmacological or surgical treatments for disease improvement, which are not entirely effective. Consequently, the lack of an efficient therapy for NAFLD continues to be a significant problem in healthcare. The current NAFLD research aims to develop single-agent and effective therapies to improve patient compliance.

As the research on intestinal microecology advances rapidly, probiotics and prebiotics have attracted increasing interest among researchers, due to their regulatory functions in the gut. Synbiotics, a synergistic blend of probiotics and prebiotics, offer an indirect yet profound advantage to human wellbeing. They operate by fostering the growth of advantageous gut microbiota, and overseeing their metabolic processes. This intricate interplay culminates in a fortified immunity, and refined metabolic equilibrium. Moreover, synbiotics have been found to improve blood glucose and insulin levels [7], which aligns with the prevailing view that NAFLD has a connection with metabolic levels and insulin resistance. This suggests the potential efficacy of synbiotics in treating NAFLD and related disorders.

This article presents a systematic review of the research on synbiotics development, intestinal microecology, NAFLD treatment, and the application of synbiotics in NAFLD treatment. Additionally, it proposes an improved synbiotic combination, comprising lactulose, arabinose, and *Lactobacillus plantarum*, which enhances the therapeutic efficacy of synbiotics for NAFLD when paired with traditional Chinese herbal *Sonchus brachyotus DC.* extracts.

## 2. NAFLD and Its Treatment

### 2.1. Nonalcoholic Fatty Liver Disease (NAFLD)

Nonalcoholic fatty liver disease (NAFLD) is a prevalent liver metabolic syndrome characterized by an abnormal metabolism and excessive lipid accumulation, resulting in steatosis in over 5% of liver cells [8]. While its pathological alterations bear a resemblance to alcoholic liver disease, NAFLD manifests in individuals who do not have a history of significant alcohol intake. It is frequently associated with metabolically abnormal obesity, featuring dyslipidemia and hyperglycemia. NAFLD is a leading cause of chronic liver diseases, progressing to nonalcoholic steatohepatitis (NASH), cirrhosis, and complications such as an abnormal glucose tolerance, hypertension, hyperviscosity, and coronary heart disease. Additionally, NAFLD has a close connection with the development of hepatocellular carcinoma [9]. Recent studies have introduced the term metabolic dysfunction-associated fatty liver disease (MAFLD) to encompass NAFLD, expanding its pathology classification, and explicitly associating it with type 2 diabetes and metabolic dysfunction. This redefinition opens up new avenues for future NAFLD treatment [10,11].

### 2.2. The Pathogenesis of NAFLD 

The pathogenesis of NAFLD is still not understood. Petersen et al. [12] suggest that NAFLD results from an overabundance of triglyceride accumulation in the liver due to an energy surplus, which is consistent with the higher prevalence of NAFLD in obese individuals. The liver accumulates lipids by absorbing fatty acids released from peripheral adipose tissue and ingested orally. Skeletal muscle insulin resistance leads to excessive hepatic fat accumulation via the shifting of glucose from skeletal muscle glycogen synthesis to de novo lipid synthesis [13]. Consequently, hepatic insulin resistance hinders the function of glycogen synthase, redirecting glucose into lipogenic processes, and promoting NAFLD. Studies using mice lacking hepatic glycogen synthase have shown increased hepatic adipogenesis and NAFLD development due to hepatic insulin resistance [14]. In conclusion, insulin resistance promotes excessive fatty acid intake and de novo lipogenesis in the liver, ultimately leading to hepatocyte dysfunction and NAFLD development.

The “second hit” theory, first proposed internationally in 1998, has gained wide acceptance [14]. It suggests that unhealthy lifestyles and dietary habits lead to lipid accumulation in the liver, causing hepatocyte steatosis and sustained cellular damage (“first hit”). This damage triggers the secretion of inflammatory cytokines, contributing to mitochondrial dysfunction, oxidative stress, and massive liver cell death, ultimately resulting in hepatocyte damage and steatohepatitis (“second hit”) (Figure 1). The emergence of NAFLD is intricately linked to heightened oxidative stress, an undue accumulation of lipids within the liver, and inflammatory processes. Research has elucidated that disruptions in lipid metabolism lead to the buildup of hepatic lipids, thereby exerting a profound impact on the generation of various reactive oxygen species (ROS). These sources of ROS span from mitochondria and the endoplasmic reticulum (ER) to NADPH oxidase, contributing to the oxidative milieu within the liver. NAFLD results in excessive mitochondrial ROS production in the liver, and increased ROS generation is involved in the regulation of insulin signaling and lipid-metabolism-related enzyme expression and activity, further promoting NAFLD development [15]. The redox signaling pathway interacts with the immune signaling network to regulate the inflammatory response. Consequently, the pathogenic progression of NAFLD is extremely intricate, prompting the formulation of the “multiple hit” theory to elucidate its developmental mechanisms [16].

Although the exact pathogenesis of NAFLD remains enigmatic, there is a consensus regarding its substantial correlation with metabolic disorders and insulin resistance. This connection is substantiated by the “second hit” and “multiple hit” theories, along with the corresponding empirical findings [17,18]. Amongst the two theories, the former garners greater acceptance. This theory posits that, subsequent to hepatocytes undergoing the initial impact, insulin resistance and leptin resistance ensue, culminating in hepatic steatosis. This process, in turn, triggers inflammation, necrosis, and fibrosis as a response to oxidative stress. This highlights the role of an abnormal endocrine axis function in NAFLD development. These findings may provide new directions for the development of drugs or functional foods to treat NAFLD, with the goal of improving insulin resistance, and preventing the second blow to hepatocytes from oxidative stress, thus preventing the development of hepatocellular carcinoma, cirrhosis, and complications.

### 2.3. The Treatment for NAFLD

Ganesh et al. [6] summarized the current global treatment options and recommendations for NAFLD in their review. Currently, there are no approved treatments for NAFLD, and weight loss interventions may be the most effective approach. Pharmacological interventions for NAFLD primarily seek to enhance insulin sensitivity, while reducing hepatic inflammation and fibrosis biomarkers. An ongoing study in the United States for NASH treatment has shown that omega-3 fatty acid esters (eicosapentaenoic acid) may be a potential candidate for the first-line treatment of hypertriglyceridemia in NAFLD patients. Certain studies propose that polyunsaturated fatty acids (PUFAs) play a pivotal role in enhancing hepatic steatosis and the biochemical markers associated with non-alcoholic fatty liver disease (NAFLD), while also improving insulin sensitivity, and mitigating inflammation [19,20]. However, another study indicated that PUFA did not have disease-modifying effects in NASH patients with diabetes [21], suggesting certain limitations in its use for these patients. Further research is required to determine the optimal dosage of omega-3 PUFA supplementation, and its effects on hepatic lipids.

Insulin sensitizers, extensively evaluated and recognized in previous treatment studies for NASH, have shown promising results. For instance, pioglitazone, a trial drug, has demonstrated an improvement in steatohepatitis, compared to vitamin E and the placebo [22]. Long-term pioglitazone treatment resulted in significant improvements in liver damage. However, long-term administration may carry the risks of cardiovascular disease and bone loss [23,24,25,26,27,28]. Metformin, commonly used to treat type 2 diabetes, also improves NAFLD, by increasing glucose utilization in the peripheral tissues. Studies have demonstrated that metformin improves insulin sensitivity, liver histology, and serum alanine transaminase (ALT) levels, particularly in obese NASH patients [29]. Vitamin E, a lipophilic antioxidant, has been shown to slow down the progression of NAFLD, by effectively reducing oxidative damage, and inhibiting inflammatory cytokine production in the liver [22,30]. Clinical trials have revealed that, after 96 weeks of treatment, a notable 43% of all participants exhibited marked histological enhancements. These improvements were particularly pronounced in relation to the amelioration of inflammation in both the hepatocyte ballooning and lobular areas. These positive changes can be largely attributed to a reduction in oxidative stress-induced damage [31].

Probiotics have also been mentioned in treatment protocols, and have shown potential benefits in NAFLD treatment. In a review published in 2021, Yao et al. [32] analyzed the application of intestinal flora in NAFLD treatment, specifically highlighting the positive effects of *Lactobacillus plantarum* in reducing ALT and AST levels in patients. The use of active ingredients from Chinese herbal medicine, following the discovery of artemisinin for the treatment of malaria, has gained increasing attention. Traditional Chinese medicine often refers to fatty liver as “liver accumulation” and “liver gangrene”, and several herbs, such as *Rhizoma Coptidis* (Huang Lian), *Radix Salvia Miltiorrhizae* (Dan Shen), *Rhei Rhizoma* (Da Huang), *Fructus Gardeniae* (Zhi Zi), and *Semen Cassiae* (Jue Ming Zi), are commonly used in NAFLD treatment, due to their natural ingredients that can reverse steatosis, regulate blood lipids, inhibit inflammation, and resist oxidative stress. Studies have shown the therapeutic effects of *Radix Salvia Miltiorrhizae* extract salvia phenolic acid B, *Semen Cassiae*, and *Fructus Gardeniae* extracts in NAFLD via the inhibition of inflammation, and anti-oxidative stress [33,34,35]. These findings provide valuable insights into, and theoretical support for, combining synbiotics with active ingredients from herbal medicines in NAFLD treatment.

## 3. Definition and Function of, and Research into, Synbiotics

### 3.1. Definition

Synbiotics are a combination of probiotics and prebiotics that promote the physiological activity of probiotics, selectively adjust the distribution of intestinal flora, and enhance the effectiveness and longevity of beneficial bacteria. Probiotics are beneficial intestinal bacteria that colonize the human body, and positively impact the body’s microecology. They achieve this goal by regulating the host mucosa and systemic immune function, or by balancing the intestinal flora, promoting nutrient absorption, and maintaining intestinal and overall health. Common examples of intestinal probiotics include *Bifidobacterium* and *Lactobacillus*. On the contrary, prebiotics are organic compounds that cannot be digested or absorbed. They are specifically designed to nourish and promote growth in the beneficial microbes essential for metabolic functions and overall development. Their profound impact on the host’s wellbeing stems from their ability to modulate changes within the intestinal microbe community. These changes, in turn, exert influence over the endocrine, barrier, and immune functions, effectively restoring a harmonious balance to the intestinal microecology. The concept of synbiotics emerged in 1995, and they were defined as a synergistic blend of probiotics and prebiotics. This unique combination serves to enhance the host’s health, by fostering the survival and successful integration of microorganism food supplements in the digestive tract. Additionally, synbiotics selectively stimulate growth in specific health-promoting bacteria, and activate their metabolism, thereby yielding a targeted and amplified positive impact [36]. This definition is consistent with those of prebiotics and probiotics, confirming their complementary and synergistic relationship. In August 2020, the International Scientific Association of Probiotics and Prebiotics (ISAPP) updated the term “synbiotics”, and redefined it as “a mixture comprising live microorganisms and substrate (s) selectively utilized by host microorganisms that confers a health benefit on the host” [37], emphasizing the valuable effects and potential of synbiotics for human health, and providing insights and guidance for future synbiotics development.

### 3.2. Functions of Synbiotics

Probiotics, functioning as beneficial microorganisms within the intestinal tract, play a pivotal role in mitigating the host’s weight, and ameliorating metabolic irregularities. This is achieved through the regulation of distinct metabolic pathways and immune responses, via their metabolites. Furthermore, they contribute to weight reduction in the host, by facilitating the absorption of fats and cholesterol. Probiotics play a role in improvements in the digestion of lactose [38], the treatment of antibiotic-associated diarrhea [39], and the prevention of necrotizing small intestinal colitis in preterm neonates, as demonstrated in systematic reviews and meta-analyses [40], remission induction in inflammatory bowel disease [41], the prevention and control of hyperglycemia [7], the improvement of lipid levels and inflammation, including a reduction in total cholesterol, the level of high-density lipoprotein (HDL), and the inflammatory marker tumor necrosis factor (TNF)-α [42], and blood glucose and insulin resistance in diabetic patients [43]. Probiotics additionally generate short-chain fatty acids. These compounds help reduce the local pH, encourage the production of immunomodulatory cytokines, and stimulate mucin generation. As a result, they foster a more robust and beneficial intestinal microenvironment [44].

Synbiotics represent a synergy between probiotics and prebiotics, endowing the host with augmented advantages. This synergy manifests through the selective enhancement of favorable bacterial growth within the intestinal tract, as well as the activation of pathways that bolster the metabolism of bacteria conducive to human wellbeing. Furthermore, they amplify the viability in, and integration of, probiotics within the gut, consequently fostering a more propitious microbial milieu within the intestines, thus promoting human health. These benefits include enhancing immunity, preventing respiratory and gastrointestinal infections, promoting lactose digestion, alleviating symptoms such as bloating and abdominal pain due to lactose intolerance, enhancing nutrient absorption, and facilitating the degradation of indigestible plant fibers into short-chain fatty acids. Overall, synbiotics significantly improve insulin secretion and sensitivity, by modulating the intestinal flora’s composition. This therapeutic effect has been demonstrated in NAFLD, via the alleviation of insulin resistance and the promotion of the organism’s metabolism. Furthermore, synbiotics enhance the absorption of antioxidant components in the combination, leading to hepatoprotective effects.

### 3.3. Gut Microecology and NAFLD

The human gut microbiota comprises 10–100 trillion microorganisms, surpassing the number of human cells. Its distribution and composition are closely related to the host’s life, and play essential roles in the host’s immune response, food digestion, regulation of the intestinal secretory function, neural signaling, drug metabolism, and catabolism [45]. In recent years, the advancements in intestinal microecology have garnered considerable interest. Studies have identified a correlation between the distribution of intestinal microflora and various diseases, leading to research on the potential treatment of human diseases through regulation of the intestinal microecology.

The “hepatic–intestinal axis” is a critical component of the human metabolic system. It entails intricate biliary, portal, and systemic interconnections, linking the liver and the intestine. This axis is regulated by various signals from the intestinal flora and its metabolites, environmental toxins, and food antigens [46]. Long-term studies have demonstrated that a healthy intestinal flora helps regulate the function of crucial metabolic organs such as the liver, promoting host metabolic stability [45]. Irregularities within the intestinal microbiota, including an excessive proliferation of pathogenic bacteria, bacterial translocation, the generation of harmful metabolites, and perturbation in the signaling for substance and energy metabolism along the gut–liver axis, have the potential to modify the immune landscape of the host. These alterations, in turn, play a pivotal role in fostering the emergence of liver-related disorders, notably non-alcoholic fatty liver disease (NAFLD) and alcoholic liver disease (ALD) [47] (Figure 2). Studies with NAFLD case studies have shown statistically significant differences in the levels of *Lactobacillus*, *Bifidobacterium*, and *Escherichia coli* in patients, compared to the control group [48]. Multiple case studies focusing on NAFLD (non-alcoholic fatty liver disease) and NASH (non-alcoholic steatohepatitis) have unveiled noteworthy distinctions in the intestinal flora, within both animal models and patients. These disparities underscore the paramount importance of the intestinal microecology in the progression of NAFLD [49,50]. Recent research indicates that interventions aimed at manipulating the gut microbiota could present a promising therapeutic avenue for liver diseases [51,52].

Moreover, NAFLD is not a singular fatty liver disorder but, rather, a spectrum of metabolism-related diseases closely associated with diabetes mellitus and metabolic syndrome. T2DM is a common complication of NAFLD, and shares similarities with NAFLD, such as insulin resistance and a chronic low-grade inflammatory state. Research has shown a close association between the intestinal microecology and T2DM, revealing potential mechanisms through which regulating the intestinal microecology benefits T2DM patients. Furthermore, patients with NAFLD and T2DM exhibit similar patterns in microflora distribution, suggesting the possibility of simultaneously improving NAFLD and T2DM by modulating intestinal microflora [53]. In light of these findings, a new synbiotic combination consisting of lactulose, arabinose, and *Lactobacillus plantarum* has been developed, showing promising results in treating T2DM [54]. Its efficacy on NAFLD model mice has been validated, and further formulation modification is underway, to obtain a new synbiotic mixture.

## 4. *Sonchus brachyotus DC.*

According to Flora of China (FRPS), both *Sonchus brachyotus DC.* and *Sonchus arvensis L.* are annual herbs belonging to the genus *Sonchus*, in the family *Compositae*. *Sonchus brachyotus DC.* is also known as *Ixeris sonchifolia*, wild bitter cabbage, and *Ixeris polycephala Cass*. It has a lengthy history of medicinal use in Chinese medicine, where it is referred to as bitter cabbage. Early Chinese medical texts, such as the classic *Shennong Threshes the Hundred Grasses* and *Compendium of Materia Medica*, describe its bitter and cold taste, and highlight its effects in clearing heat, eliminating toxins, reducing swelling and pus, promoting blood circulation and removing blood stasis, clearing the lungs and relieving cough, invigorating the liver and promoting diuresis, and aiding digestion and harmonizing the stomach. Additionally, as well as its medicinal uses, *Sonchus brachyotus DC.* is commonly consumed as a wild vegetable in folk traditions, due to its high nutritional value, as it contains protein, various amino acids, carotene, and rich trace elements [55].

Several studies have showcased that disruptions in the ecosystem of the gastrointestinal tract can hinder the organism’s typical metabolic functions. This can trigger a range of pathological damages, and contribute to the advancement of chronic diseases [56,57,58,59]. Hence, the precise manipulation of the intestinal microbiota emerges as indispensable in addressing metabolic irregularities. Over recent years, bioactive compounds derived from botanical sources have assumed a pivotal role in fostering diverse functionalities, particularly in governing the dynamics of intestinal microecology, a domain where investigations have borne remarkable outcomes. Abundant research underscores the fact that *Sonchus* species, apart from their culinary use, boast a plethora of medicinal properties [60,61,62,63,64]. *Sonchus* species boast an abundance of vitamin C and omega-3 PUFA [65], potentially imbuing them with noteworthy anti-inflammatory and antioxidant properties. Omega-3 PUFA has been employed in numerous clinical trials for the therapeutic intervention of NAFLD, according to reports [66,67,68]. Significantly, species of *Sonchus* have been documented for their potential therapeutic effects in diabetes management [65]. The findings showcased in these studies underscore the potential of *Sonchus* species in contributing to the management of metabolic syndromes. With *Sonchus brachyotus DC*. serving as a traditional vernacular herb, renowned for its efficacy in dispelling heat-related pollutants, arresting hemorrhages, and addressing ailments such as acute dysentery and enteritis, its significant anti-inflammatory and antioxidant attributes have been documented. These has been established through the identification of compounds that exhibit robust antioxidant and anti-inflammatory properties [61,69]. The components derived from *Sonchus brachyotus DC*. exhibit the ability to effectively restrain the production of free radicals, as well as the process of lipid peroxidation [61]. Furthermore, recent research has unveiled the finding that *Sonchus brachyotus DC*. harbors a plethora of bioactive constituents, notably flavonoids [70]. These compounds have demonstrated a remarkable capability to impede the proliferation of various microbes, such as *Escherichia coli*, *Enterobacter cloacae*, and *Klebsiella pneumoniae*, while concurrently fostering the induction of apoptosis [71]. Apart from its demonstration of antioxidant and antibacterial activities in vitro, there are reports suggesting that *Sonchus brachyotus DC*. holds the capability to mitigate drug-induced oxidative stress damage, by regulating the abundance of the intestinal microflora [72]. This further substantiates the favorable contribution of *Sonchus brachyotus DC*. in the treatment of NAFLD.

*Sonchus brachyotus DC.* has identified several chemical constituents in the plant, including β-sitosterol, lignan, apigenin, and quercetin, which possess biological activities, such as antibacterial, hepatoprotective, anti-inflammatory, antioxidant, and insulin-resistance-improving properties. For example, studies conducted at the University of Science and Technology of Tianjin have identified potent antioxidant active components in an extract from *Sonchus brachyotus DC*., suggesting its potential application in the treatment of NAFLD, due to its remarkable hepatoprotective and antioxidant effects (unpublished data from our collaborators). Additionally, *Sonchus* species are abundant in fatty acids, such as Omega-3 PUFA [73], which is the main active ingredient in drugs under development for NAFLD treatment, further supporting its potential use in NAFLD treatment.

Utilizing these data, the article suggests a therapeutic approach for NAFLD, involving a synergistic utilization of extracts from *Sonchus brachyotus DC*. (SBE) and a synbiotics compound. An appropriate amount of SBE is added to the previously developed synbiotics, consisting of lactulose, arabinose, and *Lactobacillus plantarum*, for compounding. The efficacy of this combination is then tested in a mouse model of NAFLD, to assess the improvement in their pathology. As a result, the addition of herbal active ingredients significantly improves the liver condition of NAFLD-model mice.

## 5. Synbiotic Compounds for NAFLD Treatment

### 5.1. Synbiotic Compounds

The synbiotic compound is composed of functional sugars and *Lactobacillus plantarum*, with a cell count of live bacteria ranging from 10^8^ to 10^10^ CFU/g under deposit number CGMCC 8198. By weight percentage, the synbiotic compound consists of 30% lactulose, 30% arabinose, and 40% *Lactobacillus plantarum*. To obtain SBE, an aqueous solution of ethanol is mixed with the crude powder of *Sonchus brachyotus DC.*, followed by the ultrasonic extraction, rotary evaporation, and freeze-drying of the supernatant. The resulting SBE is used at a dosage of 2.0 g/kg per serving.

L-arabinose is beneficial for controlling glycolipid metabolism, by inhibiting the metabolic conversion of sucrose, effectively reducing blood glucose levels. It does so by inhibiting the activity of disaccharide-hydrolyzing enzymes, which leads to an increase in sucrose metabolism [74,75]. The remaining undigested sucrose is then catabolized by microorganisms in the colon, producing organic acids that inhibit fat synthesis in the liver. L-arabinose has been demonstrated to alter the abundance and diversity of the intestinal flora as a prebiotic, resulting in improvements in the body lipid rate, blood lipid levels, fasting blood glucose, glucose tolerance status, and the extent of liver damage in an animal model of metabolic syndrome [76,77].

Lactulose, a disaccharide composed of galactose and fructose, is not digested after ingestion, but is catabolized by the intestinal flora in the colon, leading to changes in the pH value in the intestinal tract [78]. This change stimulates colonic motility, relieves constipation, and regulates the physiological rhythm of the colon. The breakdown products of lactulose also inhibit the proliferation of harmful bacteria, and alter the number of dominant intestinal flora. Studies have shown that lactulose supplementation in mice fed a high-salt diet resulted in changes in the structure of the intestinal flora, and an improve in intestinal permeability [79].

*Lactobacillus plantarum* belongs to the *Lactobacillus* genus of the *phylum Firmicutes*, and is commonly used in fermented foods, due to its numerous health benefits. Experimental studies have confirmed the beneficial effects of *Lactobacillus plantarum* in decreasing blood lipid levels, and regulating the intestinal flora [80,81,82,83,84,85,86,87]. These effects include immunomodulatory properties, lower levels of ALT and AST, the inhibition of pathogenic bacteria proliferation, decreased serum cholesterol levels, and the prevention of cardiovascular diseases. *Lactobacillus plantarum* also helps maintain the balance of the intestinal microflora, promotes nutrient absorption, alleviates lactose intolerance, and inhibits the growth of tumor cells. In this experiment, the strain used is *Lactobacillus plantarum* (L. plantarum) CGMCC 8198, which has demonstrated the abilities to decrease serum cholesterol, and exhibit antioxidant activity [88].

The synbiotic compounds composed of L-arabinose, lactulose, and *Lactobacillus plantarum* have demonstrated hypoglycemic and hypolipidemic effects, as well as control of the body weight. A study by Jiang et al. [54] used this synbiotic compound in a T2DM mouse model, and observed a significant decrease in body weight and the levels of blood glucose and blood lipids. These findings provide a basis for the further application of this combination in NAFLD. Additionally, SBE is added to the list of compounds to enhance liver antioxidative capacities and further improve the body’s metabolic capacity. The combination effectively controls blood glucose and blood lipid levels, reduces body weight, decreases ALT and AST levels, and improves liver damage in NAFLD patients.

### 5.2. Animal Experiment

In preclinical experiments, an increasing body of research underscores the therapeutic potential inherent in extracts from various traditional Chinese medicines (TCMs) for addressing NAFLD. Ye et al. [89] discovered that administering luteolin to db/db mice over an 8-week period yielded notable outcomes. Through the mediation of LXR-SREBP-1c signaling, luteolin exhibited a remarkable capacity to mitigate both liver cholesterol accumulation, and the initiation of novel lipid synthesis, in the context of NAFLD. This, in turn, contributed to the promotion of hepatic steatosis attenuation. Further investigations pertaining to NAFLD demonstrated that rutin, a natural flavonoid replete with diverse biological effects, holds promise. Rutin exhibited the potential to reinstate PPAR enzyme activity, while concurrently suppressing the expression of fatty acid synthases, including FAS and ACC, in murine models of NAFLD. This dual action effectively hindered the escalation of oxidative stress levels, and facilitated an amelioration of liver disease-associated damage [90,91]. A comparable outcome was observed in another flavonoid study conducted by Panchita et al. [92], who discovered that administering quercetin-3-O-β-glucoside (Q3G) and fructooligosaccharide (FOS) to rats on a dextrin diet for 45 days resulted in a reduction in both oral glucose tolerance tests (OGTTs) and total cholesterol levels. Additionally, there was an increase in GLP-1, indicating that the combination of Q3G and FOS holds therapeutic potential for NAFLD. *Sonchus brachyotus DC*. has been found to contain multiple identifiable flavonoids. These discoveries strongly indicate that *Sonchus brachyotus DC*. may possess predictable therapeutic effects on NAFLD. In order to substantiate the efficacy of the *Sonchus brachyotus DC*. extract and synbiotics combination for NAFLD treatment, this blend was tested on an established murine NAFLD model, to evaluate its capacity for ameliorating NAFLD pathology. Upon the successful induction of the NAFLD model, groups of seven mice were analyzed for NAFLD-related markers (Table 1), and diverse interventions were implemented during the designated period. The potential therapeutic impact of *Sonchus brachyotus DC*. on NAFLD was evaluated via an assessment of its phenotypic influence, and the liver histology and plasma levels of the pertinent cytokines in the mice.

### 5.3. Efficacy of SBE in Combination of Synbiotics in Alleviating NAFLD

Following drug interventions, the mice in the model group showed elevated blood sugar and insulin levels, while the treatment groups exhibited varying degrees of control over their blood glucose and lipids, leading to significant reductions in FBG and INS levels (Figure 3B,C). This indicates that the synbiotics combination has a significant positive effect on NAFLD progression in mice, with the experimental group (NAFLD+Synb+SBE) showing the most significant improvement. Therefore, the synbiotics combination demonstrates an effective control of blood glucose and insulin levels.

The levels of total cholesterol and triglyceride content in the mice under intervention were assessed (Figure 3D and Figure 3E, respectively). The TG level was reduced in the drug intervention groups, compared to the model group (NAFLD), and the experimental group (NAFLD+SBE+Synb) showed the most significant decrease. Although the total cholesterol (TC) levels were not significantly different, each group still demonstrated some control ability compared to the model group (NAFLD). This suggests that a combination of synbiotics and SBE plays a more effective role in regulating hyperlipidemia.

Furthermore, the alanine aminotransferase (ALT) levels in the serum of the NAFLD mice were significantly reduced by the combination of extract from *Sonchus brachyotus DC*. and synbiotics, compared to the model control group and the control groups (Figure 3F). This reduction indicates a reversal in the progression of chronic liver damage due to oxidative stress.

Histological analysis of the liver further confirmed the beneficial effects of the treatment. The liver wet weight and the liver coefficient in the embodiment and proportional groups were reduced compared to the NAFLD model group. The reduction in liver tissue abnormalities was more significant in the embodiment (SBE+Synb) and proportion 2 (SBE) groups, compared to proportion 1 (Synb), suggesting a favorable role of SBE in liver protection (Table 2). All these data are from our accepted Chinese patent (no. CN2022110780786).

### 5.4. Clinical Research

The gut microbiome is considered a key factor in the development of NAFLD [59], so the regulation of the intestinal microecology is a crucial focus for future NAFLD treatments. In recent years, the clinical research on synbiotics for NAFLD treatment has shown promising results [93,94,95,96,97,98,99,100,101,102]. The selection of appropriate probiotics and prebiotics to improve NAFLD progression is becoming increasingly feasible and predictable.

One randomized controlled trial (RCT) involving 50 NAFLD patients demonstrated that a combination of *Lactobacillus reuteri* and inulin led to reduced hepatic steatosis, along with improvements in body weight, BMI, and serum uric acid levels [95]. Another study by Behrouz et al. [103], which treated 89 NAFLD patients with various probiotics, also showed significant effects on blood glucose levels, and other related biochemical markers of NAFLD [103]. Similarly, in patients with NASH, the administration of *Bifidobacterium longum* along with fructooligosaccharides resulted in significant reductions in inflammation-related factors, AST levels, and steatosis [104]. Combining probiotics with metformin in a NASH trial showed improved outcomes for liver injury [105]. A separate RCT involving *Lactobacillus acidophilus* supplementation in NAFLD patients demonstrated reduced ALT and AST levels [106]. Aller et al. [107] utilized *Lactobacillus bulgaricus* and *Streptococcus thermophiles*, observing similar beneficial effects in NAFLD patients.

These clinical studies provide strong evidence of the positive effects of synbiotics in NAFLD treatment. Building on these developments, the positive impact of prebiotics in NAFLD treatment is further confirmed. Our experiments demonstrated that administering appropriate prebiotic combinations to model mice improved liver damage and overall condition. The intestinal microecology’s application in treating various diseases has become a significant research area, with ongoing investigations in experimental and clinical research. Compared to traditional drug therapies, synbiotics offer milder, safer, and more cost-effective options that are suitable for most patients, and have a more noticeable effect. Introducing synbiotics formally into NAFLD treatment, and enhancing their therapeutic effect through new synbiotic combinations, will offer great hope to patients.

Based on our results and models of previous clinical studies, we propose a clinical research plan for new synbiotic combinations. The plan involves several main areas of focus, including the effects of an oral intake of synbiotic combinations with SBE on the control of body weight, blood glucose levels, and other physiological indexes in NAFLD patients, and whether this treatment would effectively improve liver damage in these patients. Only systematic medical examinations would be able to reveal the clinical significance of this synbiotic and SBE combination.

## 6. Possible Mechanisms of Synbiotics in NAFLD

Numerous studies have shown that the dysregulation of the gut microbiome is strongly linked to NAFLD severity [108,109,110,111,112]. Animal experiments by Roy et al. [113] demonstrated that gut microbial transplantation (FMT) from different mouse models led to germ-free mice exhibiting distinct phenotypes of liposynthesis and steatosis. Another experiment by Chiu et al. [93] involved transferring gut microbes from NASH patients to germ-free mice on a high-fat diet, resulting in mice showing biochemical changes similar to those of NASH patients, including increased serum ALT and AST levels [114]. This suggests that the gut microbiome plays a decisive role in NAFLD development.

Further research indicates that the gut microbiome mainly influences the human body through its metabolites, with changes in bile acid metabolism signals, intestinal permeability, and short-chain fatty acid production being the main mechanisms involved [115,116,117]. Primary bile acids support fat-soluble substance digestion and absorption in the intestine, maintain the intestinal barrier, and regulate lipid and sugar metabolism, by activating receptors such as the farnesoid X receptor (FXR) and G protein-coupled bile acid receptor 1 (GPBAR1/TGR5) [118,119,120]. The activation of FXR promotes the secretion of fibroblast growth factor 15/19 (FGF15/19), which targets the liver, and inhibits hepatic accumulation and degeneration [97,98,99]. TGR5 activation increases the secretion of glucagon-like peptide-1 (GLP-1), which, together with FXR, reduces blood lipid levels, and improves fatty liver symptoms [121,122]. Intestinal flora regulation also impacts intestinal permeability, with changes in the intestinal bacteria affecting the synthesis and secretion of angiopoietin-like 4 protein (ANGPTL4) in the small intestine, thereby regulating liver fat storage [123,124].

Studies by Okubo et al. [125], using methionine-choline-deficient NASH mice, demonstrated that administering the *Lactobacillus casei* strain Shirota (Lcs) increased the number of other lactic acid bacteria. This led to decreased intestinal inflammation and serum LPS concentration in NASH mice, improved liver damage, and proved the effectiveness of the *Lactobacillus casei* strain Shirota (Lcs) in NAFLD. Similar results were obtained by Nguyen et al. [126] using *Lactobacillus plantarum* PH04, which showed significant cholesterol and triglyceride-lowering effects. A report by Zhao et al. [127] indicated that administering *Lactobacillus plantarum* NA136 to a high-fat diet-induced NAFLD mouse model effectively reduced NAFLD severity, and reversed insulin resistance (Figure 4).

## 7. Future Prospective

NAFLD is a widespread chronic liver disease affecting nearly one-quarter of the world’s population, and its prevalence is projected to keep increasing at a high rate, imposing a significant burden on healthcare systems. Currently, there are no approved treatments for NAFLD, leading to considerable distress among patients. Lifestyle modifications are considered the most reliable approach to managing the condition. However, with the deepening research on the intestinal microbiome and its connection to metabolic diseases, synbiotics, as food items capable of regulating the intestinal flora, offer great promise in alleviating NAFLD.

Various probiotics and prebiotics possess distinct characteristics, and their combination as a synbiotic provides a dietary option that benefits human health, and shows promise in regulating and alleviating diseases. This approach is expected to become one of the essential methods for preventing and treating metabolism-related diseases in the future. The synbiotic combination used in this experiment effectively regulates the intestinal flora through L-arabinose, lactulose, and *Lactobacillus plantarum*. It leads to reduced blood glucose and blood lipids, improved insulin resistance and liver damage in patients, and enhanced protection and improvement of liver function, through the addition of active ingredient extracts from traditional Chinese medicinal herbs. Animal experiments have demonstrated the positive effects of this synbiotic combination in treating NAFLD.

We anticipate that this synbiotic combination will perform similarly well in clinical trials, serving as a dietary aid to control patients’ weight, reduce blood glucose, lipid, ALT, and AST levels, improve liver damage, and promote overall health in NAFLD patients, and individuals with metabolism-related diseases. Furthermore, it may enhance the defenses of healthy individuals, and serve as a preventive measure against metabolic syndrome, NAFLD, T2DM, and other diseases, when incorporated into their diets.

Given the ongoing rise in NAFLD incidence, effective preventive or early intervention methods will significantly alleviate the burden of NAFLD on the general population. We are optimistic about the potential of the synbiotics combination research to eventually provide a low-cost, safe, and highly effective therapeutic approach to the public, bringing good news to NAFLD patients, and contributing to improved overall health.

## Figures and Tables

**Figure 1 foods-12-03393-f001:**
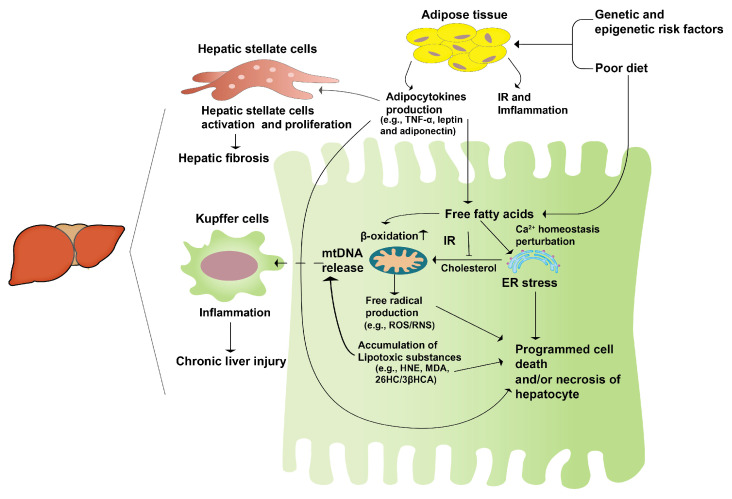
The second-hit pathogenesis of NAFLD. A genetic predisposition coupled with an unhealthy diet leads to an excessive accumulation of fatty acids in the liver. Concurrently, the heightened insulin resistance in the adipose tissue initiates an inflammatory response, prompting the release of adipocytokines. These cytokines activate hepatic stellate cells, initiating a cascade that culminates in hepatic fibrosis and fatty liver, constituting the “first hit”. Subsequently, this sets the stage for persistent hepatocyte damage, stress responses, and hepatocyte apoptosis, collectively manifesting as liver hepatitis, or the “second hit”.

**Figure 2 foods-12-03393-f002:**
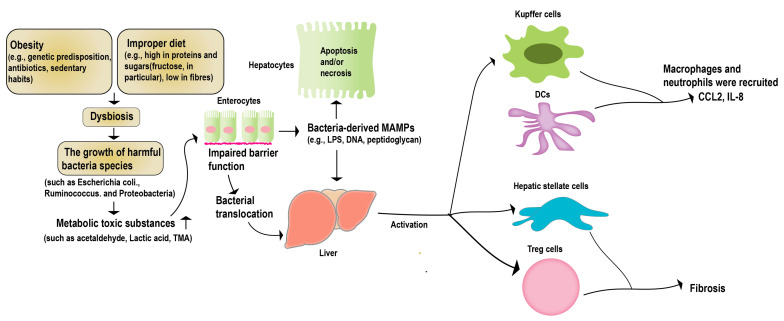
Gut microbiota and NAFLD. Obesity or a high-fat diet can trigger an imbalance in the intestinal microecology, and disrupt the function of the intestinal barrier. This disruption allows pathogenic bacteria and harmful metabolites to migrate to the liver. Consequently, this can lead to apoptosis in the liver parenchymal cells, and the activation of non-parenchymal immune cells within the liver. These cascading effects ultimately give rise to liver lesions and the progression of NAFLD.

**Figure 3 foods-12-03393-f003:**
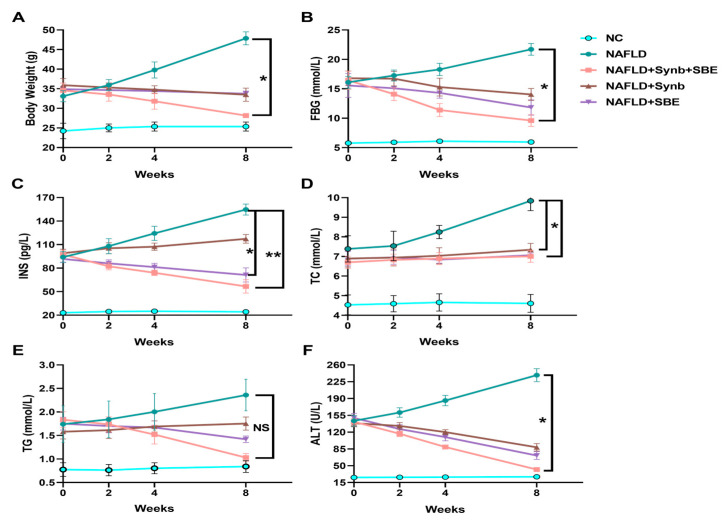
Effects of a synbiotic and SBE compound on NAFLD. After administration in mice, the changes in (**A**) body weight, (**B**) fasting blood glucose (FBG), (**C**) serum insulin concentration (INS), (**D**) total cholesterol (TC), (**E**) triglycerides (TG), and (**F**) alanine aminotransferase are shown. The representative results from three independent experiments are included. NS, no significance; * *p* < 0.05; ** *p* < 0.01. n = 3 biological replicates/group; one-way ANOVA. All analyses were performed using the GraphPad Prism 8 software (Version 8.0.2; GraphPad Software, Inc., San Diego, CA, USA).

**Figure 4 foods-12-03393-f004:**
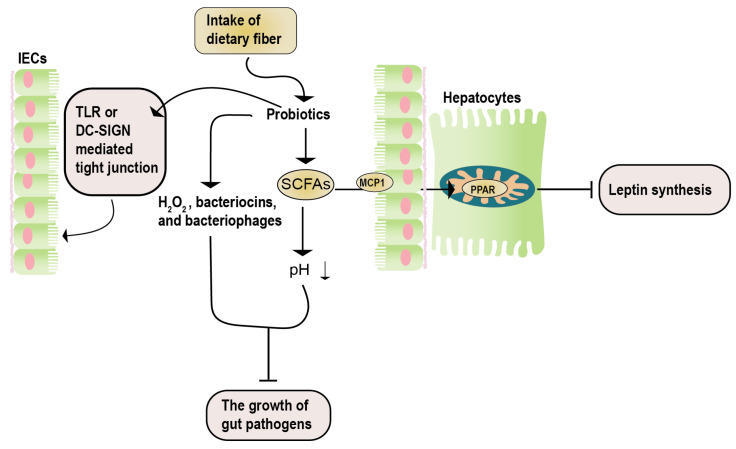
The potential mechanisms of synbiotics in NAFLD. The therapeutic impacts of synbiotics on NAFLD encompass three key aspects. Firstly, the catabolism of prebiotics results in the generation of SCFAs. These SCFAs enter hepatocytes via the MCP1 transporter and, subsequently, bind to PPAR, effectively inhibiting leptin signaling. Secondly, the catabolism of prebiotics yields active acidic substances (such as H_2_O_2_ and SCFAs), along with antibacterial agents (including bacteriocin and bacteriophages). This dual action serves to impede the proliferation of detrimental bacteria. Lastly, the influence of probiotics is manifested in their ability to enhance the maintenance of the intestinal barrier integrity. This is achieved through mechanisms involving the TLR- or DN-SIGN-mediated regulation of tight junctions. DC-SIGN, dendritic cell-specific intercellular adhesion molecule-3-grabbing non-integrin; IECs, intestinal epithelial cells; MCP1, monocyte chemoattractant protein 1; SCFAs, short-chain fatty acids; TLR, toll-like receptor.

**Table 1 foods-12-03393-t001:** The proportion of symbiotics components and supplementation.

Groups Assignment	Symbiotic Components			Supplementation
Experimental Group	30% Lactulose	30% Arabinose	40% *Lactobacillus plantarum* CGMCC 8198	2.0 g/kg SBE
Control Group 1	30% Lactulose	30% Arabinose	40% *Lactobacillus plantarum* CGMCC 8198	\
Control Group 2	\	\	\	2.0 g/kg SBE

**Table 2 foods-12-03393-t002:** Results of the liver histological analysis.

Groups	Liver Wet Weight	Liver Coefficient
Blank Group (NC)	1.137	0.0433
Model Group (NAFLD)	2.274	0.0477
Experimental Group (NAFLD+Synb+SBE)	1.347	0.0426
Control Group 1 (NAFLD+Synb)	1.663	0.0481
Control Group 2 (NAFLD+SBE)	1.408	0.0447

## Data Availability

The data are available from the corresponding author.

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
