# Peer review of "Benefits of Combining Sonchus brachyotus DC. Extracts and Synbiotics in Alleviating Non-Alcoholic Fatty Liver Disease"

_foods, 2023, doi:10.3390/foods12183393_

Round 1

Reviewer 1 Report

Huang et al., in the article entitled “Benefit of Combining Sonchus brachyotus DC. Extracts and Synbiotics in Alleviating Non-Alcoholic Fatty Liver Disease” have reviewed literature about NAFLD, its pathogenesis, and some treatment options and supported the findings of others through their own experiments on mouse models. It is an interesting and enjoyable article; however, some general and particular comments are given below to consider to improve the quality of the manuscript for readers.  

1.       Line 34: Put a full stop after the reference number. For example, instead of  “……and mortality. [2]”, it should be “……and mortality [2].” This correction needs to be made throughout the article.

2.       Line 60: “This paper” could be changed to “This article”.

3.       Line 63: Make specie and organism’s names italic. “Lactobacillus Plantarum”. This correction needs to be made throughout the article.

4.       Line 82: Petersen et al.,…… should be “Petersen et al., [12]…… Add the reference number with the name of the authors as you have done in line 433. Make this correction throughout the article.

5.       Line 132: An ongong study…… is an “ongoing study”

6.       Lines 192-193: check the definition of synbiotics “a mixture of microorganisms and…….”

7.       Line 198: reducing host’s weight

8.       Line 256: pattern of microflora distribution

9.       Line 258: Instead of “new prebiotic combination…” it should be “a new symbiotic combination….” because it contains compound and microorganisms.

10.   Figure 2 does not match the text explanation given in lines 247-248. Improve the figure by mentioning the gut-liver axis and some metabolites production which could be missing during gut dysbiosis (right side of the figure).

11.   The left side of Figure 2 has “Health” which does not make sense. The spelling “diat” should be “diet” in Figure 2 (right side).

12.   Lines 280-283: Sonchus brachyotus DC. has identified? This sentence needs to be rephrased to remove grammar mistakes.

13.   Line 290-291: “Based on this information, the paper proposes a combination of Sonchus brachyotus” extracts (SBE) and a synbiotic compound. For what??? This is an incomplete sentence so should be completed.

14.   Line 300: “30% lactose” or “30% lactulose”?

15.   Lines 307-308: “which leads to a decrease in sucrose intake” could be “which leads to a decrease in sucrose metabolism”?? By inhibiting the activity of disaccharide-hydrolyzing enzymes, sucrose metabolism would be inhibited instead of its intake.

16.   Line 344: To verify the efficacy of symbiotic mixture for NAFLD treatment……

17.   Lines 366-369: Mention the values of “significant improvement”  as compared to the controls in addition to referring to the figures.

18.   Lines 396-397: “In recent years, clinical research on synbiotics for NAFLD treatment has shown promising results”. Give the reference.

19.   Why authors used 2g/kg of extract? What is the maximum amount of crude extract that could be used as a supplement? Mention if the active ingredient which has hepatoprotective or antioxidant activity is known.

20.   English editing to remove some general grammar mistakes and rephrasing of some sentences is required.

21.   The authors have used animal models but ethical approval is missing.

General English editing to remove some general grammar mistakes and rephrasing of some sentences is required.

Author Response

Point 1: Put a full stop after the reference number. (Line 34)

Response1: We were really sorry for our careless mistakes. We have carefully checked and adjusted the format of all reference citations within the article with your suggestions. (Line 37)

Point 2: “This paper” could be changed to “This article”. (Line 60)

Response 2: We were really sorry for our careless mistakes and have made changes to all of them. (Line 62)

Point 3: Make specie and organism’s names italic. “Lactobacillus Plantarum”. (Line 63)

Response 3: We were really sorry for our careless mistakes. We have amended the font format here (Line 65) and adjusted the scientific names of species throughout the article in line with your suggestions.

Point 4: “Petersen et al.,……” should be “Petersen et al., [12]……” (Line 82)

Response 4: We were really sorry for our careless mistakes and have modified the citation in accordance with your suggestion. (Line 84)

Point 5: An ongong study…… is an “ongoing study” (Line 132)

Response 5: We were really sorry for our careless mistakes and have made changes to this mistake. (Line 140)

Point 6: Check the definition of synbiotics “a mixture of microorganisms and…….” (Lines 192-193)

Response 6: We were really sorry for our careless mistakes and have supplemented the definition of this concept with a search of literature. (Line 205-207)

Point 7: reducing host’s weight. (Line 198)

Response 7: We were really sorry for our careless mistakes and have corrected this misspelling. (Line 211)

Point 8: pattern of microflora distribution. (Line 256)

Response 8: We were really sorry for our careless mistakes. The improper word has been updated according to your suggestion. (Line 275)

Point 9: Instead of “new prebiotic combination…” it should be “a new symbiotic combination….” because it contains compound and microorganisms.(Line 258)

Response 9: We were really sorry for our careless mistakes and have modified the description in accordance with your recommendation. (Line 281)

Point 10: Figure 2 does not match the text explanation given in lines 247-248. Improve the figure by mentioning the gut-liver axis and some metabolites production which could be missing during gut dysbiosis (right side of the figure).

Response 10: Thanks for the suggestion; we made changes to Figure 2 and added textual explanations to make it more reader-friendly. (Line 254-260)

Point 11: The left side of Figure 2 has “Health” which does not make sense. The spelling “diat” should be “diet” in Figure 2 (right side).

Response 11: Thanks for the suggestion. As mentioned in Response 10, we have modified this figure to make it more reader-friendly. (Line 282)

Point 12: Sonchus brachyotus DC. has identified? This sentence needs to be rephrased to remove grammar mistakes. (Lines 280-283)

Response 12: Thank you for your proposal, the active components of Sonchus brachyotus DC. were identified through experimental methodology in our previous studies, and currently the data has not yet been published in the form of a research paper but rather through a legal patent.

Point 13: “Based on this information, the paper proposes a combination of Sonchus brachyotus” extracts (SBE) and a synbiotics compound. For what??? This is an incomplete sentence so should be completed. (Line 290-291)

Response 13: We were really sorry for our careless mistakes and have added and modified this text in order to make it a complete sentence. (Line 344-345)

Point 14: “30% lactose” or “30% lactulose”? (Line 300)

Response 14: We were really sorry for our careless mistakes. This is a misspelling, which has been corrected to lactulose. (Line 355)

Point 15: “which leads to a decrease in sucrose intake” could be “which leads to a decrease in sucrose metabolism”?? By inhibiting the activity of disaccharide-hydrolyzing enzymes, sucrose metabolism would be inhibited instead of its intake. (Lines 307-308)

Response 15: We were really sorry for our careless mistakes and have changed the sentence as you suggested. (Line 362-363)

Point 16: To verify the efficacy of symbiotic mixture for NAFLD treatment…… (Line 344)

Response 16: We were really sorry for our careless mistakes. This sentence is misrepresented and has been addressed in the text for reader understanding. (Line 417-418)

Point 17: Mention the values of “significant improvement” as compared to the controls in addition to referring to the figures. (Lines 366-369)

Response 17: We were really sorry for our careless mistakes. The Figure 3 has been redrawn based on statistical analyses of the original data using statistical software, and all relevant information has been included in the annotations.

Point 18: “In recent years, clinical research on synbiotics for NAFLD treatment has shown promising results”. Give the reference. (Lines 396-397)

Response 18: We were really sorry for our careless mistakes and had included related citations in the text. (Line 465-466)

Point 19: Why authors used 2g/kg of extract? What is the maximum amount of crude extract that could be used as a supplement? Mention if the active ingredient which has hepatoprotective or antioxidant activity is known.

Response 19: Thank you for your proposal. As mentioned in Response 12, the component and activity of SBE was characterised in our previous studies and drug dosage experiments determined 2 kg/mg to be the optimal dose. The experimental methods can not be described in detail here at present.

Point 20: English editing to remove some general grammar mistakes and rephrasing of some sentences is required.

Response 20: Thank you for your helpful advice. We are currently debating and editing the whole article in terms of sentence structure and grammar so as not to confuse the readers.

Point 21: The authors have used animal models but ethical approval is missing.

Response 21: We were really sorry for our careless mistakes. We will include proof of ethical approval with the revised draft.

Reviewer 2 Report

The very high worldwide prevalence and severity of non-alcoholic fatty liver disease (NAFLD) urges researchers to uncover, characterize, and validate novel compounds with hepatoprotective properties. In the present manuscript, the authors test the conjunction of Sonchus brachyotus DC. (a Chinese herb) with synbiotics, demonstrating enhanced hepatoprotective effects.

After background information on the pathophysiology of NAFLD and the challenges regarding its treatment, the Sonchus brachyotus DC. Chinese herb, and specific synbiotic compounds for NAFLD treatment are introduced. Then, the authors describe animal experiments showing additive beneficial effects of Sonchus brachyotus DC. and synbiotics. Furthermore, they discuss these results in the framework of related clinical studies and potential physiological mechanisms of synbiotics in NAFLD.

The present manuscript contains original, unpublished results (sections 5.2-5.3), which is not permitted in review articles. The authors should re-write the manuscript as an original research paper (with minimum background information) and include a detailed methods section. From their results, the authors should introduce the innovative concept proposing that the combination of Sonchus brachyotus DC. extracts and synbiotics could enhance their beneficial effects against NAFLD. The present work could be a significant contribution to its field; however, it is not presented in the correct article format.

Moderate language editing is required, such as italics in genera and species names.

Author Response

Point 1: The present manuscript contains original, unpublished results (sections 5.2-5.3), which is not permitted in review articles. The authors should re-write the manuscript as an original research paper (with minimum background information) and include a detailed methods section.  From their results, the authors should introduce the innovative concept proposing that the combination of Sonchus brachyotus DC. extracts and synbiotics could enhance their beneficial effects against NAFLD. The present work could be a significant contribution to its field;  however, it is not presented in the correct article format.

Response1: Thank you sincerely for your valuable suggestion. The data presented in this manuscript originates from a Chinese patent application for which we have sought acceptance (Patent No. CN2022110780786). The initial research concerning this topic comprises comprehensive data encompassing drug effects, pharmacokinetics, molecular mechanisms, etc. This information will be presented in the format of a research paper submission soon. In the revision of this manuscript, we have incorporated animal findings from published studies concerning the treatment of NAFLD using plant extracts. These results are elaborated upon in the initial sections of the manuscript, from line 427 to line 461.

We wish to express our profound gratitude for your esteemed consideration of our impending revelations. It is our fervent hope that these forthcoming discoveries will enrich the existing reservoir of knowledge and effectively bridge the prevailing gaps within this specialized realm of study.

Reviewer 3 Report

This paper praises benefits of Combining Sonchus brachyotus extract on Non-Alcoholic Fatty Liver Disease which is an interesting topic. Most of the sections are written very briefly and review is also very concise and need to be elaborated a bit more for proper scrutiny of the paper. Major re-write is warranted

1.       Scientific names must be italicised throughout the manuscript including in the title.

2.       Good resolution figures must be given in the review paper and also double check copy right issue or software name should be given.

3.       How you have drawn tables? Justification must be given.

4.       References are not uniform.

Another round of review is needed after incorporation of comments given to the authors. Major revision

Author Response

Point 1: Most of the sections are written very briefly and review is also very concise and need to be elaborated a bit more for proper scrutiny of the paper.

Response1: Thank you for your proposal. We have enriched the descriptions for each section of the article. In the case of animal studies, we've diminished the reliance on subjective portrayals and instead incorporated pertinent existing research on herbal medicines concerning NAFLD. Given the broad spectrum of aspects and the similarity in mechanisms of action among several active compounds, we've selectively included a subset of studies pertaining to compounds derived from extracts of animals with NAFLD, as identified in our prior experiments. Our review, titled "Synbiotics and Gut Microbiota: Novel Insights in the Management of Type 2 Diabetes Mellitus," presents a comprehensive overview of the advancements in research related to gut microbes.

Point 2: Scientific names must be italicised throughout the manuscript including in the title.

Response 2: We were really sorry for our careless mistakes. We have edited the text and highlighted them with a distinct font style. We have edited the text and highlighted them with a distinct font style. (Displayed in red)

Point 3:Good resolution figures must be given in the review paper and also double check copy right issue or software name should be given.

Response 3: Thank you for your proposal. The software used for graphing and data analysis has been mentioned in the figure notes of Figures 1 and 3, and all images are high-resolution (300 dpi). In addition, we declare that the elements used to draw the images are not copyrighted.

Point 4:How you have drawn tables? Justification must be given.

Response 4: The table conforms to the template provided on the MDPI website, with the first row serving as the title column, the first column for the various categories, three columns for a standard three-line table, and four or more columns for a single-line table. As there is more than one person to complete the work for each section of the article, it is possible that some of the standard journal requirements were not met due to individual differences in writing or graphing styles. Currently, our team coordinates during the entire manuscript revision period and has completed the production of images and tables in accordance with the journal's layout specifications.

Point 5:References are not uniform.

Response 5: We were really sorry for our careless mistakes, We have followed the example shown in the given example to put the reference in the right place in the text --–“Author 1, A.B.; Author 2, C.D. Title of the article. Abbreviated Journal Name Year, Volume, page range.” If all contributing authors of the cited literature are more than three when we directly use the first author's name and so on in the form of “..., et al., ..." followed by the same as in the above example. And we have cited the reference to the correct location in the text.

Round 2

Reviewer 2 Report

I thank the authors for having addressed my concern. Since review papers should not rely on unpublished data, I recommend highlighting in the manuscript (as a limitation) that unpublished results were included in this study.

Author Response

Point 1: I thank the authors for having addressed my concern. Since review papers should not rely on unpublished data, I recommend highlighting in the manuscript (as a limitation) that unpublished results were included in this study.

Response1: We gratefully thanks for the precious time the reviewer spent making constructive remarks. In the most recent version of the revised manuscript, we have added and highlighted the necessary information in the section of Data Availability Statement. (Line 583-584)

Reviewer 3 Report

It is fine now.

Acceptable

Author Response

Point 1: It is fine now.

Response1: We gratefully thanks for the precious time the reviewer spent making feedback. Thank you again for your efforts during this time, and we wish you a joyful life!
